# Myosin Transducer Inter-Strand Communication Is Critical for Normal ATPase Activity and Myofibril Structure

**DOI:** 10.3390/biology11081137

**Published:** 2022-07-29

**Authors:** William A. Kronert, Karen H. Hsu, Aditi Madan, Floyd Sarsoza, Anthony Cammarato, Sanford I. Bernstein

**Affiliations:** 1Department of Biology, Molecular Biology Institute, Heart Institute, San Diego State University, San Diego, CA 92182, USA; wkronert@sdsu.edu (W.A.K.); khsu@sdsu.edu (K.H.H.); sarsoza@yahoo.com (F.S.); 2Department of Medicine, Division of Cardiology, Johns Hopkins University, Baltimore, MD 21205, USA; amadan@fredhutch.org (A.M.); acammar3@jhmi.edu (A.C.)

**Keywords:** myosin, *Drosophila melanogaster*, ATPase, transducer, myofibril, hypertrophic cardiomyopathy

## Abstract

**Simple Summary:**

Myosin is a molecular motor protein that is critical for using stored energy to yield contraction in both skeletal muscle and in the heart. Mutations in myosin can result in inherited human hypertrophic cardiomyopathy, wherein the heart walls become thickened. This results in inability to adequately pump enough oxygenated blood to meet bodily demands, and can lead to abnormal heart beating patterns and the possibility of sudden death. Typically, over-active myosin can be causative for this disease. Here, we used the common fruit fly to study one of these cardiomyopathy-causing mutations by expressing the mutant protein in the indirect flight muscle of the fly. Surprisingly, we found that the mutation and associated mutations decrease myosin and muscle function instead of enhancing it. We show that at least some of these defects result from improper interactions within the myosin protein. This suggests that appropriate communications within myosin are critical for its function and further indicates that certain aspects of indirect flight muscle myosin are functionally distinct from those of myosin in the human heart. By understanding the intricate interactions within the myosin motor protein that are important for its normal role, therapeutics for improving mutant myosin function can be developed.

**Abstract:**

The R249Q mutation in human β-cardiac myosin results in hypertrophic cardiomyopathy. We previously showed that inserting this mutation into *Drosophila melanogaster* indirect flight muscle myosin yields mechanical and locomotory defects. Here, we use transgenic *Drosophila* mutants to demonstrate that residue R249 serves as a critical communication link within myosin that controls both ATPase activity and myofibril integrity. R249 is located on a β-strand of the central transducer of myosin, and our molecular modeling shows that it interacts via a salt bridge with D262 on the adjacent β-strand. We find that disrupting this interaction via R249Q, R249D or D262R mutations reduces basal and actin-activated ATPase activity, actin in vitro motility and flight muscle function. Further, the R249D mutation dramatically affects myofibril assembly, yielding abnormalities in sarcomere lengths, increased Z-line thickness and split myofibrils. These defects are exacerbated during aging. Re-establishing the β-strand interaction via a R249D/D262R double mutation restores both basal ATPase activity and myofibril assembly, indicating that these properties are dependent upon transducer inter-strand communication. Thus, the transducer plays an important role in myosin function and myofibril architecture.

## 1. Introduction

Myosin II is an ATP-dependent molecular motor of muscle. Myosin-containing thick filaments are interdigitated with actin-containing thin filaments in repeating sarcomeric units of muscle myofibrils. The myosin heads act as cross-bridges during the actomyosin mechanochemical cycle. Rotation of the myosin lever arm in attached cross-bridges produces the force necessary for muscle contraction (reviewed by [1]). During the mechanochemical cycle, communication between myosin’s actin interface and its nucleotide-binding pocket is facilitated by a structure known as the transducer, which is composed of a seven-stranded β-sheet and its associated linker and loop domains [2]. The transducer was first identified in non-muscle myosin V, where its distortion was posited to permit sequential release of the products of ATP hydrolysis that accompanies lever arm movement and force generation [2]. It is through the transducer’s twisting that switch I of myosin’s upper 50-KDa domain and the P-loop of the N-terminal domain are repositioned to release nucleotide, a process that is also coordinated by surface loop 1 of the transducer [3,4]. 

Mutation R249Q in strand 6 of the transducer’s seven-stranded β-sheet in β-cardiac myosin II leads to hypertrophic cardiomyopathy [5,6,7]. This mutation results in left-ventricular hypertrophy with varying degrees of disease severity. Previously, we found that expressing the R249Q transducer mutation in the indirect flight muscle (IFM) of the fruit fly *Drosophila melanogaster* disables flight due to decreased muscle kinetics and power production arising from a reduced number of cross-bridges bound to actin [8].

Here, we examine the in vitro properties of full-length R249Q IFM myosin and report reductions in actin-activated ATPase activity and in vitro actin filament motility, suggesting that the mutation yields an under-active myosin. This contrasts with the enhanced activity of human β-cardiac R249Q myosin, which is proposed to arise from disruption of the interacting heads motif of the myosin dimer, wherein the molecule generally does not hydrolyze ATP rapidly or interact with actin [9,10,11]. Further, we determine that a more severe mutation, R249D, also reduces ATPase and in vitro motility and produces abnormal IFM myofibrils. Molecular modeling shows that both R249 mutations disrupt interactions with D262 on the adjacent strand of the β-sheet of the transducer. Interestingly, we find that D262R, a putative suppressor mutation of R249D, improves both basal ATPase activity and myofibril assembly in the double mutant, indicating that inter-strand communication is critical to these properties. 

## 2. Materials and Methods

### 2.1. Protein Structure Modeling

The near-rigor human β-cardiac myosin crystal structure (PDB: 4P7H) was used as a template to analyze the predicted protein structure of the R249 and D262 mutants. The wild-type indirect flight muscle isoform (IFI) myosin S1 amino acid sequence as well as versions containing R249D, D262R or R249D-D262R were modeled onto the human β-cardiac myosin backbone using the SWISS-MODEL homology modeling server (http://swissmodel.expasy.org/; accessed on 15 May 2013) [12]). Structures were viewed using the PyMOL Molecular Graphics System v. 1.8 (Schrödinger, LLC).

### 2.2. DNA Constructs

Site-directed mutagenesis was used to introduce a series of mutations (R249Q, R249D, D262R and R249D-D262R) into a *Drosophila P* element-containing *Mhc* transgene. Construction of R249Q was previously described [8]. For production of R249D, D262R and R249D-D262R, wild-type genomic construct pwMhc2 [13] was digested with *Eag*I to produce two subclones, pwMhc-5′ and pMhc-3′. The pwMhc-5′ subclone contains an 11.3-kb *Eag*I *Mhc* fragment that was cloned into the pCasper vector [14], and the pMhc-3′ subclone contains a 12.5-kb *Eag*I *Mhc* fragment that was cloned into the pBluescriptKS *Eag*I site (Stratagene, La Jolla, CA, USA). The pwMhc-5′ subclone was further digested with *Xho*I and *Avr*II. A 6.8 kb *Xho*I-*Avr*II-digested fragment from pwMhc-5′ was gel isolated and ligated into an *Xho*I-*Avr*II site in the pLitmus 28I vector (New England Biolabs, Ipswich, MA, USA), to produce the pMhc-5′-XA subclone. This was digested with *Pst*I and *Avr*II. A 4.3 kb *Pst*I-*Avr*II fragment was gel isolated and ligated into a *Pst*I-*Avr*II site in vector pLitmus 28I, to produce the pMhc-5′-PA subclone. This was further digested with *Age*I and *Bam*HI, and the resulting 370 bp fragment was gel isolated and ligated into an *Age*I-*Bam*H I site in the pLitmus 28I vector to produce the pMhc-5′-AB370 subclone. This subclone was subjected to site-directed mutagenesis using the QuickChange II kit (Agilent Technologies, Santa Clara, CA, USA). For R249D, the primer 5′- CAAGGGTAAATTCATCGACATCCACTTCGGACCC -3′ containing the R249D nucleotide mutation (underlined) was used to yield pR249D-5′AB370. For D262R, the primer 5′- TAAACTGGCTGGTGCTCGTATTGAGACCTGTAAGT -3′ containing the D262R nucleotide mutation (underlined) was used to yield pD262R-5′AB370. For the R249D-D262R construct, the pR249D-5′AB370 clone was used as the template, and the primer 5′- TAAACTGGCTGGTGCTCGTATTGAGACCTGTAAGT -3′ containing the D262R nucleotide mutation (underlined) was employed to yield pR249D-D262R-5′AB370. Upon sequence confirmation of the R249D, D262R and R249D-D262R site-directed mutagenesis products, the pR249D-5′AB370, pD262R-5′AB370 and pR249D-D262R-5′AB370 subclones were separately and sequentially cloned back into the intermediate subclones from which they originated. The resulting subclones, pwMhcR249D-5′, pwMhcD262R-5′ and pwMhcR249D-D262R-5′ were digested with *Eag*I. The 12.5-kb pMhc-3′ fragment was digested with *Eag*I, gel isolated and ligated into the *Eag*I site of each of these subclones to yield pwMhcR249D, pwMhcD262R and pwMhcR249D-D262R. The entire coding region and all ligation sites of the final plasmids were confirmed by DNA sequencing (Eton Bioscience Inc., San Diego, CA, USA) before *P* element transformation.

### 2.3. P Element Transformation of Mhc Genes

Transgenic *Drosophila* lines for pwMhcR249D, pwMhcD262R and pwMhcR249D-D262R were generated using *P* element-mediated germline transformation [15] by BestGene, Inc. (Chino Hills, CA, USA). Twelve hundred embryos were injected for each construct. Transgenes for each resulting transgenic line were mapped to their chromosome locations using balancer chromosomes and standard genetic crosses. For pwMhcR249D, thirteen transgenic lines were obtained. Seven inserts mapped to the second chromosome and six to the third chromosome. For pwMhcD262R, nine transgenic lines were obtained. Five inserts mapped to the second chromosome, one to the X chromosome and three to the third chromosome. For pwMhcR249D-D262R, three transgenic lines were obtained. Two inserts mapped to the third chromosome and one to the fourth chromosome. Two or more independent transgenic lines for each construct that mapped to the third, fourth or X chromosome were examined in detail. These were *pwMhcR249D-6* and *pwMhcR249D-12*; *pwMhcD262R-1*, *pwMhcD262R-6* and *pwMhcD262R-7*; *pwMhcR249D-D262R-2*, *pwMhcR249D-D262R-4* and *pwMhcR249D-D262R-5.* Each of these transgenes were crossed into the *Mhc*^10^ background that eliminates endogenous myosin expression in the IFM and jump muscles [16]. 

### 2.4. Reverse-Transcription Polymerase Chain Reaction (RT-PCR)

RT-PCR was used to confirm that the *Mhc* transcripts from transgenic lines were spliced correctly and contained the appropriate site-directed nucleotide changes. RNA was prepared from upper thoraces of 2-day-old female flies using the LiCl_2_ extraction method [17]. The Protoscript cDNA synthesis kit (New England Biolabs) was used to generate cDNAs for each transgenic line with a reverse specific primer (5′- TCGAACGCAGAGTGGTCAT -3′) to exon 10 and a forward specific primer (5′-TGGATCCCCGACGAGAAGGA-3′) to exon 2. The resulting RT-PCR products were sequenced by Eton Bioscience, Inc. San Diego, CA, USA.

### 2.5. Determination of Myosin Expression Levels

To determine myosin expression levels relative to actin accumulation for each transgenic line, sodium dodecyl sulphate polyacrylamide gel electrophoresis (SDS-PAGE) was used as previously described [18]. Briefly, five replicates of upper thoraces from six, 2-day-old homozygous female flies were homogenized in 60 µL SDS gel buffer, and 6 µL of each homogenate was loaded on a 9% polyacrylamide gel. Coomassie blue stained gels were digitally scanned and protein accumulation levels were determined and analyzed using NIH Image J software (https://imagej.nih.gov/ij/ accessed on 1 June 2013). Mean value ± SEM of myosin/actin ratio for each transgenic line relative to *pwMhc2* wild-type control are reported. 

### 2.6. Flight Testing

More than 100 flies of each genotype were assessed for flight ability in small groups (typically 10–15 flies) at either two or seven days post-eclosion. Flies were released into a Plexiglas box with a light at the top [19] and classed as to their flight trajectory: upward (U), horizontal (H), downward (D) or not at all (N). The flight index for each cohort was calculated as 6U/T + 4H/T + 2D/T + 0N/T, where T is the total number of flies per cohort [20]. The average flight index ± SEM is reported for each genotype and age. 

### 2.7. Confocal Imaging of Hemithoraces and IFM Myofibrils

Two-day-old flies were anesthetized using FlyNap (Carolina), and heads and abdomens were removed. Thoraces were fixed overnight in 4% formaldehyde at 4 °C and rinsed in 1× Phosphate Buffered Saline (PBS, pH 7.4: 5.60 mM Na_2_HPO_4_, 1.06 mM KH_2_PO_4_, 154.0 mM NaCl) the next day (3 × 15 min, room temperature). Specimens were laid supine on a glass slide and snap-frozen in liquid nitrogen for 10 s. Frozen thoraces were immediately bisected down the mid-sagittal plane using a razor blade. Hemithoraces were stained with mouse anti-α-actinin antibody (1:50 in 1× PBS-T: PBS with 0.1% Triton-X 100) overnight at 4 °C. Samples were rinsed in 1× PBS the next day (3 × 15 min, room temperature), before being incubated with a mixture of secondary antibody (1:10,000 Alexa-488 goat anti-mouse, Invitrogen) and Alexa-568 phalloidin (1:50 in 1× PBS-T, Invitrogen) for 2 h at room temperature. Samples were rinsed in 1× PBS (3 × 5 min, room temperature) before being mounted on a glass slide with Vectashield (Vector Laboratories) and viewed using a 10× air lens to assess gross muscle morphology, and a 100× oil immersion lens (1.25 NA) to assess myofibrillar morphology on a Leica TCS SPE RGBV confocal microscope (Leica Microsystems, Buffalo Grove, IL, USA).

### 2.8. Transmission Electron Microscopy

Transmission electron microscopy was performed as previously described [21]. Four developmental stages were assayed: late stage pupal, two-hour-old adult, two-day-old adult and seven-day-old adult. Sections were obtained from females for at least three different flies per transgenic line.

### 2.9. ATPase and In Vitro Motility Assays

For steady-state ATPase and in vitro motility assays, myosin was isolated from ~150 dissected sets of dorsolongitudinal IFMs by fiber permeabilization, high salt extraction, low salt precipitation, removal of residual actomyosin and an additional low salt precipitation, as previously described [22,23,24]. Young flies (3–20 h post-eclosion) were utilized for myosin isolation to ensure the protein had not been subject to degradation during aging. G-actin was purified from acetone powder of chicken breast muscle and polymerized to produce F-actin [23,25].

Steady-state Mg-ATPase assays were carried out using 2 µg of myosin, [γ-^32^P]-ATP and 0–2 µM of F-actin as previously described [22,23,24]. Reactions were run in duplicate for 25 min at room temperature. Following quenching with HClO_4_ and organic extraction, radioactive inorganic phosphate was assayed by scintillation counting. Prism (GraphPad Software, San Diego, CA, USA) was used to graph phosphate levels (after subtracting basal activity) vs. actin concentration to yield a Michaelis–Menten plot. *V_max_* and *K_m_* values were calculated using the Michaelis–Menten equation and are expressed as mean ± SD. Values among genotypes were compared using Welch’s one-way ANOVA followed by Dunnett’s T3 multiple comparisons test. Statistical significance was assumed for *p* < 0.05. 

In vitro motility analyses were carried out in the presence of ATP using fluorescently labeled F-actin interacting with myosin bound to nitrocellulose-coated coverslips, as previously described [22,23]. Video images were captured under fluorescence optics via a high-speed video camera linked to a computer. Following capture using Piper Software (Stanford Photonics, Burlingame, CA, USA), filament velocity was analyzed using NIH ImageJ. At least 30 smoothly moving filaments were assessed per myosin sample. Values are presented as mean ± SD. Values for all genotypes were compared using Welch’s one-way ANOVA followed by Dunnett’s T3 multiple comparisons test. Statistical significance was assumed for *p* < 0.05.

## 3. Results

### 3.1. Molecular Modeling of R249 Interactions

The R249 residue is critical to myosin function based on the observations that the R249Q mutation causes human cardiac disease [5,6,7] and *Drosophila* muscle dysfunction [8]. Our molecular modeling previously showed that R249 forms a salt bridge with residue D262, linking strands 6 and 7 of the transducer’s β-sheet [8]. To study the effects of mutations in these residues, we modeled the *Drosophila* IFM myosin heavy chain sequence onto the human β-cardiac myosin motor domain (PDB: 4P7H), as shown in Figure 1A. Note that the R249 and D262 residues and the general structure of the transducer are conserved between human and *Drosophila* myosins [26]. The wild-type myosin model shows that positively charged R249 residue interacts with negatively charged residue D262 with a 2.6 Å contact distance (Figure 1B). The mutation R249D reverses the charge of the interacting moiety, disrupting the salt bridge and increasing the contact distance with D262 to 3.8 Å (Figure 1C). Likewise, the charge reversal mutant D262R disrupts the salt bridge and increases the contact distance with R249 to 4.2 Å (Figure 1D). We found that the R249D-D262R double mutation is predicted to reestablish a charge attraction between the residues, producing a 2.9 Å contact distance (Figure 1E). Hence, each single mutation is expected to disrupt the local interaction between strands 6 and 7 of the β-sheet, while the double mutation should restore the linkage. We next proceeded to test the effects of these mutations in *Drosophila melanogaster* transgenic flies, which allow biochemical, structural and functional dissection of myosin mutation phenotypes [13]. 

### 3.2. Production and Verification of R249D, D262R and R249D-D262R Transgenic Lines

We used in vitro mutagenesis to introduce *R249D*, *D262R* and *R249D-D262R* mutations into the single *Drosophila* muscle *Mhc* gene. DNA sequence analysis confirmed the correct nucleotide base changes for each DNA construct. We then employed *P* element-mediated transformation to stably insert the transgenes into the *Drosophila* germline and mapped the chromosomal location of each insertion. At least two independent transgenic lines for each construct that mapped to the third, fourth or X chromosome were examined in detail. Transgenic lines that mapped to the second chromosome were not studied since the endogenous myosin gene is located on that chromosome. Each transgene was crossed into the *Mhc*^10^ background, which eliminates endogenous myosin expression only in the indirect flight and jump muscles [16]. RT-PCR of a 2.5 kb cDNA from upper thoraces of each line and subsequent DNA sequencing confirmed the presence of the mutant transgene and appropriate RNA splicing, which includes the expected alternative exons surrounding the transgene mutations (exons 3b, 7d and 9a). 

To determine myosin expression levels for the transgenic lines, we extracted proteins from upper thoraces of two-day-old adult female flies. SDS-PAGE and densitometric analysis allowed for determination of the ratio of myosin heavy chain to actin levels. The ratio of expression of myosin:actin for each transgenic line was essentially equal to wild-type levels (Table 1).

### 3.3. Effects of the R249D, D262R and R249D-D262R Mutations on Flight Ability

We determined flight ability for each of the homozygous transgenic lines in an *Mhc*^10^ background (Table 2). For two-day-old and seven-day-old adult females, flight ability of the *pwMhcR249D*, *pwMhcD262R* and *pwMhcR249D-D262R* was severely impaired compared to the control wild-type transgenic line *pwMhc2*. While the control flight index averaged in the “up” to “horizontal” range (4.6 and 4.1 at days 2 and 7, respectively), the mutant lines yielded mean values of 0 to 1.7, with flies displaying mostly downward flight or no flight at all (Table 2). Notably, the R249D mutation completely disabled flight (Table 2), compared to the milder R249Q mutation that yielded a flight of ~1.8 [8]. Further, while the D262R mutation allowed some flight (flight index of ~1.6 and ~0.8, at 2 and 7 days, respectively), combining this mutation with R249D did not rescue flight ability (0.3 and 0 indexes at 2 and 7 days for R249D-D262R, respectively). 

### 3.4. Effects of the R249D, D262R and R249D-D262R Mutations on Muscle Ultrastructure

We examined whether gross level defects exist in structures of *R249D*, *D262R* and *R249D-D262R* IFM that could affect muscle function. To this end, we bisected thoraces from late-stage pupae, two-hour-old adults, two-day-old adults and seven-day-old adults for each line and stained for actin with fluorescent phalloidin. In the oldest adults examined, the structure of dorsal longitudinal IFMs from the mutants and *pwMhc2* controls showed normal size and shape of fibers running from anterior to posterior (Figure 2), as had been seen previously in IFMs of R249Q [8]. No hypercontraction of fibers was evident, in contrast to IFMs from some stocks with aberrant actin-myosin interactions that yield excess force [27]. However, for R249D only, occasional thinning of myofibrils resulted in gaps within the fiber structures (Figure 2B). 

We examined sarcomere structure in the same fibers via confocal microscopy at each of the four developmental stages by staining for actin and for alpha-actinin, which localizes to Z-lines (Figure 3). While myofibrils from *pwMhc2* wild-type control (Figure 3A–D) and *D262R* (Figure 3I–L) appeared normal, *R249D* myofibrils showed several irregularities (Figure 3E–H). These included myofibril branching (magenta boxes), thick Z-line-like structures that stain intensely with phalloidin (cyan arrowheads), and occasionally both (yellow arrowheads). These defects became more abundant and pronounced with age. Interestingly, myofibrillar defects observed via confocal microscopy were not evident in the *R249D-D262R* double-mutant myofibrils (Figure 3M–P) until 7 days of age, where we observed some myofibrillar damage (purple boxes), suggesting (at this level of resolution) that the D262R residue is able to suppress the early structural defects engendered by the R249D mutation.

We next utilized transmission electron microscopy to determine effects of the mutations on myofibril structure and stability at high resolution (Figure 4). Sections of the *pwMhc2* transgenic control appeared wild-type throughout development (Figure 4A–D). In transverse sections, myofibril morphology remained rounded during aging, with standard thick and thin filament double-hexagonal packing; in longitudinal sections, Z- and M-lines are normally arrayed to form sarcomeric structures. In contrast, *R249D* organisms show myofibril assembly defects that are exacerbated during aging (Figure 4E–H). Late-stage pupae and two-hour-old adults display assembly defects including gaps in myofilaments and sarcomere bifurcations (Figure 4E,F, magenta boxes). Two-hour-old *R249D* adults exhibit apparent compaction of myofilament arrays (Figure 4F, top). Two-day-old and seven-day-old *R249D* adults continue to display progressive disruption in muscle structure with further compaction of filaments, myofibrillar gaps, and myofibril branching (Figure 4G,H: magenta box), as well as apparent misplaced Z-line material/shortened sarcomeres (Figure 4H, cyan arrowhead). In contrast, *D262R* transverse and longitudinal sections from late-stage pupae and two-hour-old adults resemble those of the transgenic control (Figure 4I,J). While longitudinal sections from two-day-old and seven-day-old *D262R* adults display relatively normal sarcomeres, except for some gaps in filament packing, transverse sections show disruptions in thick and thin filament hexagonal packing (Figure 4K,L). For the *R249D*-*D262R* double mutant, transverse and longitudinal sections from late-stage pupae and two-hour-old adults (Figure 4M,N) resemble those of the transgenic control. This is significantly improved compared to late-stage pupae and two-hour-old adults of *R249D* (Figure 4E,F). Transverse and longitudinal sections from two-day-old and seven-day-old *R249D*-*D262R* adults (Figure 4O,P) are also improved compared to *R249D* (Figure 4G,H). Longitudinal sections of the double mutant show some gaps in the filament arrays and irregularities in sarcomere width, but lack the severe gapping, branching and Z-line displacement present in *R249D.* Transverse sections show that myofibril integrity is maintained fairly well in *R249D-**D262R* two-day-old adults, except for fraying at the periphery, but this deteriorates at seven days (Figure 4O,P). Still, defects are not as severe as in *R249D* myofibrils at the same ages (Figure 4G, H), nor those in *D262R* transverse sections (Figure 4K,L). Overall, the high-resolution analysis indicates that while defects in assembly and/or maintenance of myofibrillar structure are evident in both the *R249D* and the *D262R* mutants, incorporation of both mutations into the myosin molecule yields normal myofibrillar assembly and better myofibril maintenance during aging.

### 3.5. Effects of the R249Q, R249D, D262R and R249D-D262R Mutations on Myosin ATPase and Actin Sliding Velocity

To examine the effects of the *R249D*, *D262R* and *R249D-D262R* mutations on myosin function at the biochemical/biophysical level, we measured basal and actin-activated ATPase activity as well as the ability of myosin to propel actin filaments in vitro. We also assessed the effects of the *R249Q* mutation that causes hypertrophic cardiomyopathy (HCM). A summary of the results obtained for each myosin and a listing of statistically significant differences are shown in Table 3 and Figure 5, with primary data provided in Appendix A. 

Basal Mg-ATPase activity (Table 3) was not significantly reduced for the R249Q HCM mutation compared to the wild-type control (0.25 ± 0.04 vs. 0.30 ± 0.11, respectively). However, the more severe R249D mutation as well as the D262R mutation reduced basal Mg-ATPase activity significantly relative to the control (0.12 ± 0.05 and 0.11 ± 0.04, *p* < 0.001). Interestingly, combining these two mutations in R249D-D262R myosin yielded activity that was significantly higher (0.24 ± 0.03, *p* < 0.05) than either individual mutation and not significantly different from the wild-type control. This indicates that the R249D and D262R mutations suppress each other’s defective basal Mg-ATPase activity. 

Actin-stimulated ATPase activities were significantly reduced by more than 65% for all three single-mutant myosins relative to the wild-type control (Table 3 and Figure 5A). This deficit was not improved in the R249D-D262R double mutant, indicating that the rescue of *V_max_* values does not arise from rebuilding the salt bridge between the two stands of the β-sheet of the myosin transducer. Likewise, the *K_m_* values that signify the amount of actin that yields half-maximal myosin ATPase activity were significantly reduced by more than 50% for all mutant myosins, including R249D-D262R, indicating that the double mutation does not improve this property. 

As with actin-stimulated ATPase, the velocity of actin filaments moving over myosin molecules was reduced by ~65% or more for each of the mutant myosins (1.66 ± 0.11–2.54 ± 0.25 µm/s) relative to wild-type control (7.06 ± 0.65 µm/s, *p* < 0.0001; Table 3 and Figure 5B). The R249D mutation (1.66 ± 0.11 µm/s) had the most severe effect, which was not significantly improved in the double mutant (1.97 ± 0.13 µm/s). 

Overall, actin-independent basal Mg-ATPase activity was reduced by severe disruptions of the interactions between myosin residues 249 and 262, but this was ameliorated by compensatory mutations that repair the salt bridge between these residues. In contrast, the actin-dependent parameters of *V_max_*, *K_m_* and myosin-driven actin motility did not improve from their abnormal values when the salt bridge was restored. 

## 4. Discussion

Our integrative analysis using the *Drosophila* system probed the role of myosin heavy chain residue R249 in myosin function at the biochemical, biophysical, ultrastructural and locomotory levels. This residue is of interest in that it is within the central β-sheet of the transducer that is posited to play a critical role in communication between the nucleotide pocket and the actin binding site [1,2,4]. R249 is also of importance because the R249Q mutation is causative of human hypertrophic cardiomyopathy (HCM) [5,6,7]. Thus, this study provides insight into both myosin mechanochemistry and human muscle disease. 

By introducing the R249D mutation, we destroyed a salt bridge linking the sixth strand of the central β-sheet of the transducer to residue D262 on the seventh strand (Figure 1). This results in reductions in myosin ATPase activity and in vitro actin motility (Table 3), aberrant muscle ultrastructure (Figure 3 and Figure 4) and absence of flight muscle function (Table 2). Similar deficiencies arise for the D262R mutation, which also eliminates the salt bridge, although the effects of D262R are not as severe upon muscle ultrastructure and flight ability as R249D. Interestingly, combining the two mutations, which molecular modelling predicts will restore the salt bridge (Figure 1E), returns basal ATPase activity to normal levels and improves myofibril assembly and stability. It does not, however, improve actin-activated ATPase activity, in vitro motility or flight capability. Hence, while we can conclude that transducer inter-strand communication is critical for basal ATPase activity and assembly/initial stability of myofibrils, other aspects of myosin function presumably require these residues to perform additional roles in the mechanochemical cycle beyond inter-strand communication. 

How might inter-strand communication within the transducer be necessary for building and maintaining myofibril structure? Given that the transducer is within the motor domain of myosin and its twisting serves to communicate between the nucleotide pocket and the actin binding site, it is reasonable to posit that transducer mutations could yield inappropriate interactions with the thin filament during myofibril assembly. Interactions with the thin filament may be necessary for normal myofibril stability as well, since the transducer mutations result in increased myofibrillar disarray during aging. These defects include myofibril degeneration and splitting, as well as accumulation of abnormal Z-line structures. Such abnormalities have been noted to arise in *Drosophila* from reductions in thin filament proteins [28], reduced levels of proteins required for tension generation [29], as well as from decreases in proteins that alter transcription [30] or alternative RNA splicing of myofibrillar components [31]. Interestingly, a model of human β-myosin-based HCM displays thickened Z-discs in myofibrils of stem-cell-derived cardiomyocytes [32]. It should be noted that the age-related degradation of muscle structure observed in the mutant *Drosophila* lines may arise from a process akin to disuse atrophy, where reduced force output leads to transcriptional and protein stability changes yielding myofibrillar proteolysis.

As the transducer is critical in communication between the nucleotide pocket and the more distal actin binding site and in regulating the power stroke through relayed interactions with the converter and lever arm [1,2,4], defective transducer inter-strand interaction presumably affects both the ATPase cycle and the mechanical cycle. Indeed, our muscle mechanical studies of the R249Q mutation showed a reduction in myosin cross-bridge binding to actin due to slowed attachment and enhanced detachment rates [8]. These abnormal thick and thin filament interactions are mirrored at the molecular level in reductions in actin-activated ATPase activity and actin filament in vitro motility (Table 3, Figure 5). 

While the reduced ATPase activity and in vitro motility results for HCM mutant R249Q *Drosophila* myosin are consistent with our previous fiber mechanical investigation showing decreases in actin binding and power output [8], they are at odds with studies using expressed human R249Q myosin, where increased availability of myosin heads for actin interaction was demonstrated compared to wild-type myosin [9]. Further, our results contrast with the commonly promulgated theory that such enhanced myosin function initiates HCM (reviewed in [33]). In this regard, many myosin HCM mutations, including R249Q [9,10,34], appear to disrupt the interacting heads motif (IHM), a structure in which dimeric myosin heads interact with each other (with one designated as “blocked” and one as “free”) and with the myosin S2 domain to reduce actin binding and inhibit actin-activated ATPase activity (reviewed in [35]). The IHM conformation may correspond to the super-relaxed state of muscle [36], wherein ATP-bound myosin hydrolyzes ATP at extremely slow rates [37]. The molecular model building of β-cardiac myosin IHM suggests that residue R249 stabilizes this structure by the binding of the blocked head to the free head S2 region [9,10,34] and the binding of the free head residue to the blocked head residue E409 [10]. In these models, introduction of the R249Q mutation is predicted to destabilize the energy-saving IHM, yielding higher levels of myosin activity. Note, however, that in the absence of the S2 binding site, activity of human R249Q myosin is not increased relative to wild-type myosin [9].

The failure of *Drosophila* IFM myosin to readily form the IHM motif might explain the lack of enhanced function for its R249Q mutation. Electron microscopy demonstrated that *Drosophila* IFM myosin dimers are less capable of forming the IHM compared to *Drosophila* embryonic myosin [38]. Further, ultrastructural analysis of relaxed *Drosophila* thick filaments showed myosin heads in random orientations [39], rather than in the folded IHM conformation seen in the slower IFM relaxed thick filaments of the insect *Lethocerus* [40] or in tarantula [41]. Finally, a recent cryo-electron microscopic study of relaxed *Drosophila* IFM thick filaments failed to yield a structure for the S1 domains, suggesting that they are randomly oriented rather than engaged in an ordered IHM on the thick filament backbone [40]. Notably, a protein thought to be stretchin-KLP blocks binding of the free head to the filament backbone, preventing formation of the regular IHM array seen in *Lethocerus* IFM [40]. Note that while the IHM may not be found in ordered form arising from binding on the *Drosophila* thick filament backbone, a version could still occur without backbone interaction [40]. Overall, however, there is evidence from in vitro and isolated thick filament studies that *Drosophila* IFM myosin activity may not be influenced by the standard IHM conformation, suggesting that enhanced myosin function might not be expected from the R249Q mutation in the *Drosophila* model. Future analysis in the *Drosophila* system could focus on whether embryonic myosin, which is more capable of forming the IHM [38], yields enhanced myosin function in the presence of the R249Q mutation. This differential effect would suggest that isoform-specific IHM stability is a potential mechanism for regulating myosin isoform function. In this regard, human fast myosin has been shown to have fewer heads in the super-relaxed state than slow myosin [42]. *Drosophila*, with its strong escape response and fast wing beat frequency may take this to the extreme for IFM myosin by eliminating the IHM substantially or entirely. It should also be noted that while the contractile properties of human cardiac muscle and *Drosophila* IFM have similarities [43], there are protein differences (for example, human myosin binding protein C and *Drosophila* flightin) as well as differences in cellular architecture that may play a role in the differential responses of these two muscle types to the R249Q mutation. 

## 5. Conclusions

We examined the importance of maintaining the integrity of a salt bridge connecting two strands of the β-sheet within the transducer domain of muscle myosin heavy chain in *Drosophila melanogaster*. One of these residues is causative of HCM when mutated in human β-cardiac myosin. Mutating either amino acid residue of the salt bridge had severe effects upon basal ATPase activity, actin-activated ATPase activity, actin in vitro motility, myofibril assembly and stability, and flight muscle function. Restoring the salt bridge with compensatory mutations rescued basal ATPase activity and improved myofibril assembly and stability. Thus, while the salt bridge connecting the strands of the transducer is critical for these latter parameters, additional roles played by the wild-type residues are necessary for actin-based ATPase/motility, complete myofibril stability and flight muscle function. 

## Figures and Tables

**Figure 1 biology-11-01137-f001:**
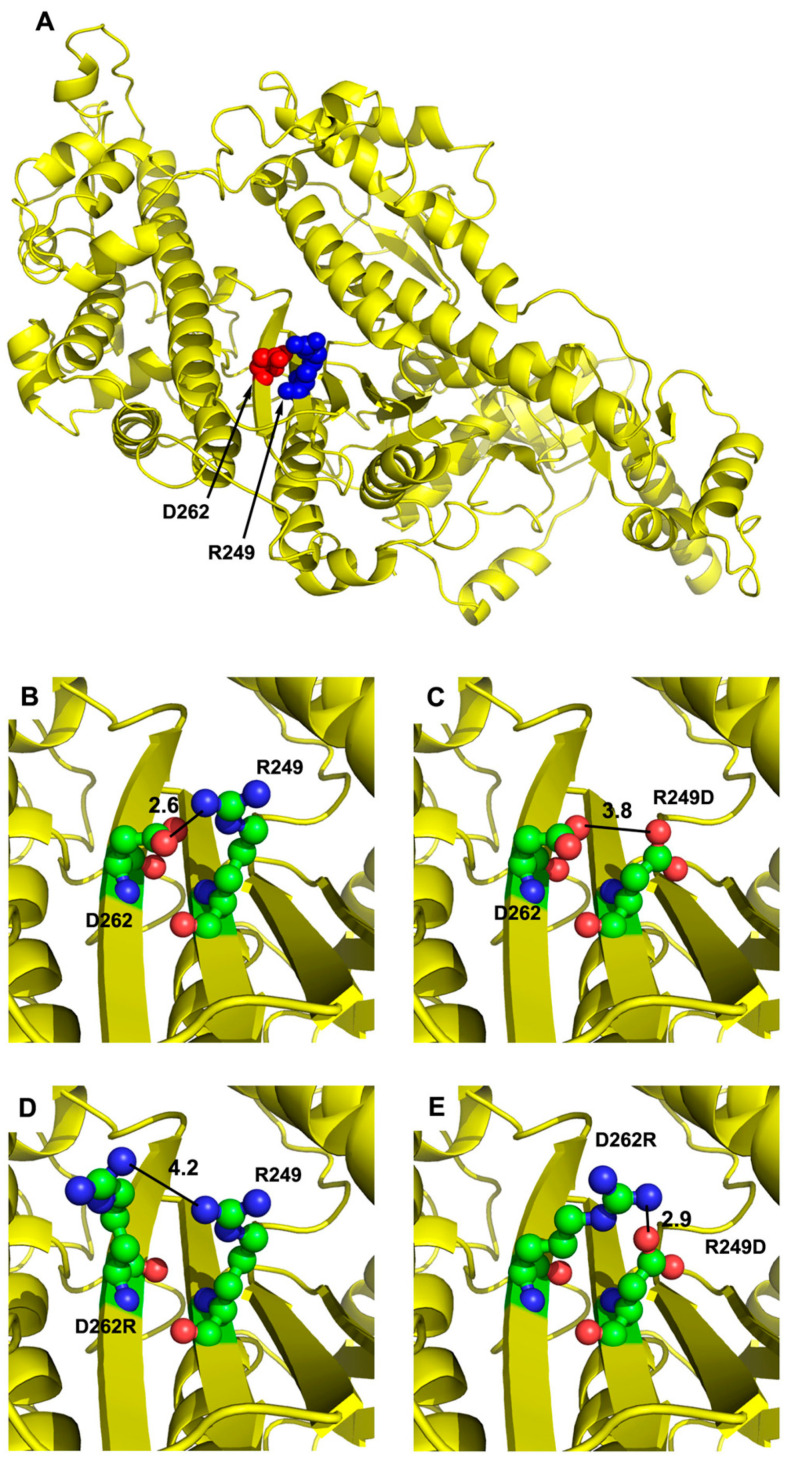
Location and interaction of transducer residues R249 and D262 on the myosin molecule. (**A**) The human β-cardiac myosin motor domain (PDB: 4P7H) was used as a template for modeling the *Drosophila* indirect flight muscle myosin heavy chain sequence to determine the location of R249 (displayed as blue spheres) and D262 (red spheres). The R249 residue is located on the sixth strand and D262 is located on the seventh strand of the central β-sheet that is a portion of the myosin transducer. (**B**) The positively charged R249 residue interacts with negatively charged D262, with a 2.6 Å contact distance. (**C**) The R249D mutation disrupts the charged interaction with D262 due to their opposing charges, yielding an increase in contact distance to 3.8 Å. (**D**) The D262R mutation disrupts the charged interaction with R249 due to their opposing charges, yielding an increase in contact distance to 4.2 Å. (**E**) The R249D-D262R double mutation reestablishes the charged interaction with a contact distance of 2.9 Å.

**Figure 2 biology-11-01137-f002:**
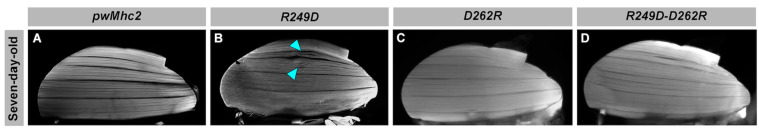
Normal gross level structure of dorsolongitudinal IFM fibers of *R249D*, *D262R* and *R249D-D262R* mutant adults. Bisected thoraces were stained with phalloidin and imaged via fluorescence microscopy. Shown are seven-day-old adults with the following genotypes: (**A**) *pwMhc2* wild-type control, (**B**) *R249D*, (**C**) *D262R*, (**D**) *R249D*-*D262R* double mutant. Fibers are of generally normal shape and size in all genotypes, indicating that there are no gross level defects, such as hypercontraction. However, *R249D* fibers show gaps (cyan arrowheads).

**Figure 3 biology-11-01137-f003:**
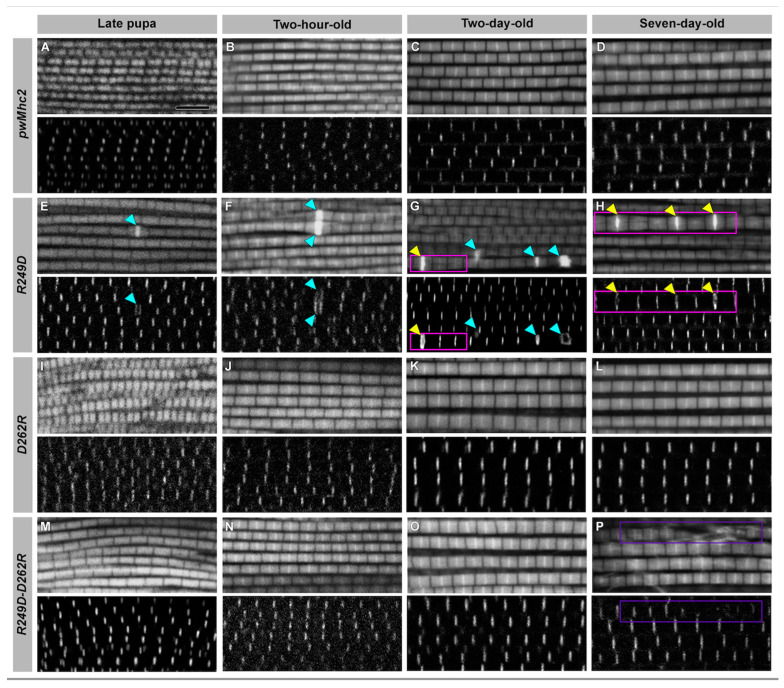
Confocal images of *R249D*, *D262R* and *R249D-D262R* myofibrils show defects in *R249D* that are rescued in the double mutant. Myofibrils from four developmental stages were imaged in longitudinal orientation to assess sarcomere structure. Muscles are stained for actin with fluorescent phalloidin and with an antibody to alpha-actinin, which localizes at Z-lines. *pwMhc2* controls display regular Z- and M-lines within sarcomeric structures throughout development: (**A**) *pwMhc2* late-stage pupa, (**B**) *pwMhc2* two-hour-old adult, (**C**) *pwMhc2* two-day-old adult, (**D**) *pwMhc2* seven-day-old adult. (**E**–**H**) Myofibrils from *pwMhcR249D* display minor assembly defects that are exacerbated during development. These include thick Z-line-like structures (cyan arrowheads), myofibril branching (magenta boxes), and occasionally both defects (yellow arrowheads). (**I**–**L**) Myofibrils from *pwMhcD262R* show essentially wild-type sarcomere structures throughout development. (**M**) Myofibrils assemble normally in the *pwMhcR249D*-*D262R* double mutant at the late-pupal stage. (**N**–**O**) Normal structure is maintained in young adults for *pwMhcR249D*-*D262R*. However, defects are present in seven-day-old *pwMhcR249D*-*D262R* myofibrils ((**P**), purple box). Thus, at this level of resolution, the *D262R* mutation rescues the various ultrastructural defects seen in *pwMhcR249D* myofibril assembly and early adult life. Scale bar, 5 µm.

**Figure 4 biology-11-01137-f004:**
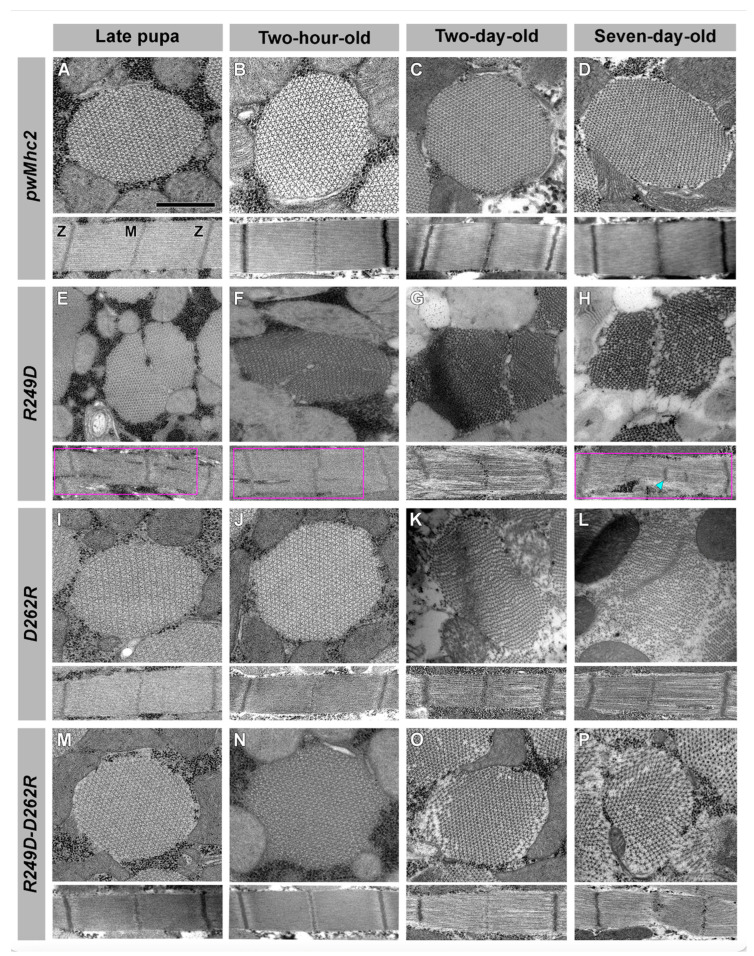
Transmission electron microscopy of *R249D*, *D262R* and *R249D-D262R* myofibrils shows severe defects in *R249D* that are largely rescued in the double mutant. Transverse and longitudinal sections from (**A**) *pwMhc2* late-stage pupae, (**B**) *pwMhc2* two-hour-old adults, (**C**) *pwMhc2* two-day-old adults, (**D**) *pwMhc2* seven-day-old adults. Transverse sections display thick and thin filaments in a normal double-hexagonal pattern throughout development and aging. Longitudinal sections show regularly spaced Z- and M-lines. (**E**) Transverse and longitudinal sections from *pwMhcR249D* late-stage pupae display assembly defects with some disruption in thick and thin filament packing. Branching of the sarcomere is seen in longitudinal section (magenta box). (**F**) Transverse and longitudinal sections from *pwMhcR249D* two-hour-old adults show similar defects to late-stage pupae. Gaps in the sarcomere are seen in longitudinal section (magenta box). (**G**) Transverse and longitudinal sections of *pwMhcR249D* two-day-old adults display further disruption in muscle structure, with filament packing, myofibril branching and disrupted sarcomeres. (**H**) Transverse and longitudinal sections from *pwMhcR249D* seven-day-old adults display abnormal myofibril structure (magenta box), branching myofibrils and irregular Z-line placement that may correspond to thickened actin-containing structures (cyan arrowhead and adjacent Z-line structure to its right). (**I**,**J**) Transverse and longitudinal sections from *pwMhcD262R* late-stage pupae and two-hour-old adults resemble those of the transgenic control (Panels A and B), with normal hexagonal packing of thick and thin filaments and regular sarcomere structures. (**K**) Transverse section from *pwMhcD262R* two-day-old adult displays disruptions in thick and thin filament hexagonal packing. Sarcomere structure in longitudinal section remains relatively intact, with some myofilament gaps. (**L**) Transverse section from *pwMhcD262R* seven-day-old adult displays further disruptions in thick and thin filament hexagonal packing, with filaments loosely dispersed throughout the myofibril. Sarcomere structure remains relatively intact in longitudinal section, with some myofilament gaps and infiltration of Z-line material. (**M**,**N**) Transverse and longitudinal sections from *pwMhcR249D-D262R* late-stage pupae and two-hour-old adults resemble those of transgenic control (Panels A and B), with normal hexagonal packing of thick and thin filaments and normal sarcomere structures. This contrasts with abnormal myofibril morphology and filament packing in late-stage pupae and two-hour-old adults of *pwMhcR249D* (Panels E and F). (**O**) *pwMhcR249D-D262R* two-day-old adults display peripheral disruptions in thick and thin filament hexagonal packing in transverse section, with relatively normal sarcomere structure in longitudinal section. Myofibril morphology is less disrupted than in two-day-old *pwMhcR249D* adults (Panel G) or *pwMhcD262R* adults (Panel K). (**P**) Transverse and longitudinal sections from *pwMhcR249D-D262R* seven-day-old adults display more widespread disruption in thick and thin filament hexagonal packing. Sarcomere structure shows some disruption of M- and Z-lines. Myofibril morphology and filament packing in seven-day-old *pwMhcR249D*-*D262R* adults are less disrupted compared to *pwMhcR249D* (panel H) or *pwMhcD262R* seven-day-old adults (Panel L). Scale bar, 0.55 µm. M, M-line; Z, Z-line.

**Figure 5 biology-11-01137-f005:**
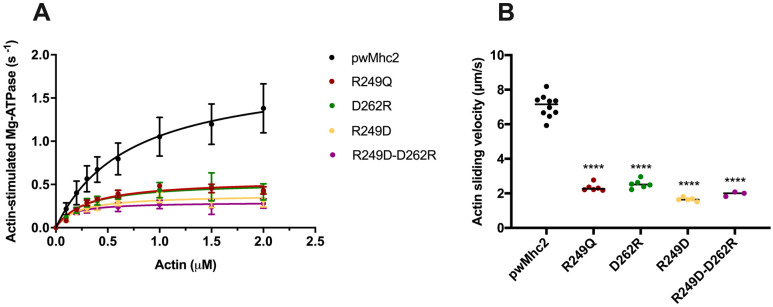
Actin-stimulated ATPase and actin sliding velocity values are reduced for R249Q, R249D, D262R and R249D-D262R myosin molecules. (**A**) Actin-stimulated Mg-ATPase levels are plotted relative to actin concentration for control (PwMhc2) and mutant myosins, after subtraction of basal Mg-ATPase values determined in the absence of actin. Each of the mutant myosins dramatically reduces Mg-ATPase activity relative to control (see Table 3
*V_max_* values) and decreases the concentration of actin needed to yield half-maximal ATPase activity (see Table 3
*K_m_* values). No statistically significant improvement is observed in myosin containing the R249D-D262R double mutation relative to the single mutations. (**B**) Actin sliding velocities calculated for control (PwMhc2) and mutant myosins. All mutations significantly decrease actin sliding velocities (****) and no improvement is observed in myosin containing the R249D-D262R double mutation relative to the single mutations (Table 3).

**Table 1 biology-11-01137-t001:** Transgenic lines, locations and myosin expression levels.

Line Name	Chromosomal Location ^a^	Protein Accumulation ± SEM ^b^
*pwMhc2*	X	1.00 ± 0.04
*pwMhcR249D-6*	3	0.98 ± 0.07
*pwMhcR249D-12*	3	0.95 ± 0.06
*pwMhcD262R-1*	3	0.96 ± 0.09
*pwMhcD262R-6*	3	0.95 ± 0.02
*pwMhcD262R-7*	3	0.97 ± 0.01
*pwMhcR249D-D262R-2*	3	0.95 ± 0.02
*pwMhcR249D-D262R-4*	4	0.95 ± 0.01
*pwMhcR249D-D262R-5*	3	0.91 ± 0.02

^a^ Chromosomal location of transgene insertions determined using balancer chromosomes and standard genetic crosses. ^b^ Two-day-old adult female transgenic flies in an *Mhc*^10^ background were assayed for myosin protein expression levels relative to actin expression levels (*n* = 5). These ratios were then divided by the mean value of *pwMhc2* samples (control). Protein accumulation represents means ± standard error of the mean (SEM). Primary protein accumulation data are given in Appendix A.

**Table 2 biology-11-01137-t002:** Flight ability of *pwMhc2*, *pwMhcR249D*, *pwMhcD262R* and *pwMhcR249D-D262R* transgenic lines.

Line Name	Fly Age (Days)	Number Tested	Up (%) ^a^	Horizontal (%)	Down (%)	Not at All (%)	Flight Index ± SEM ^b^
*pwMhc2*	2	148	54.1	24.3	17.6	4.1	4.6 ± 0.01
*pwMhcR249D-6*	2	119	0	0	0	100	0
*pwMhcR249D-12*	2	127	0	0	0	100	0
*pwMhcD262R-1*	2	130	0.8	18.5	40.8	40.0	1.6 ± 0.02
*pwMhcD262R-6*	2	121	1.7	19.0	40.5	38.8	1.7 ± 0.03
*pwMhcD262R-7*	2	121	0.8	18.2	41.3	40.0	1.6 ± 0.03
*pwMhcR249D-D262R-2*	2	128	0	0	15.6	84.4	0.3 ± 0.01
*pwMhcR249D-D262R-4*	2	132	0	0	15.2	84.8	0.3 ± 0.01
*pwMhcR249D-D262R-5*	2	130	0	0	15.4	84.6	0.3 ± 0.01
*pwMhc2*	7	116	37.1	36.2	21.6	5.2	4.1 ± 0.02
*pwMhcD262R-1*	7	131	0	0	44.3	55.7	0.88 ± 0.02
*pwMhcD262R-6*	7	131	0	0	42.7	57.3	0.85 ± 0.02
*pwMhcD262R-7*	7	117	0	0	35.9	64.1	0.72 ± 0.01
*pwMhcR249D-D262R-2*	7	118	0	0	0	100	0
*pwMhcR249D-D262R-4*	7	116	0	0	0	100	0
*pwMhcR249D-D262R-5*	7	119	0	0	0	100	0

^a^ Flight abilities expressed as percentages of the entire number of flies tested that flew up (U), horizontally (H), down (D) or not at all (N). ^b^ Flight index calculated based upon mean value ± SEM of cohorts tested (see Materials and Methods and Appendix A) using the equation 6U/T + 4H/T + 2D/T + 0N/T, with T being the total number of flies tested in each cohort [20].

**Table 3 biology-11-01137-t003:** ATPase activities and in vitro motility velocities of wild-type and mutant myosin.

Myosin Type (n for ATPase/n for Motility)	Basal Mg-ATPase (s^−1^)	Actin-Stimulated *V_max_* (s^−1^)	Actin-Stimulated *K_m_* (µM)	Actin Velocity (µm/s)
IFM control- pwMhc2 (18/10)	0.30 ± 0.11 ^a^	1.79 ± 0.41 ^d^	0.71 ± 0.23 ^e^	7.06 ± 0.65 ^f^
R249Q (6/6)	0.25 ± 0.04 ^b^	0.56 ± 0.11	0.31 ± 0.09	2.34 ± 0.22 ^g^
R249D (5/5)	0.12 ± 0.05 ^c^	0.40 ± 0.22	0.22 ± 0.17	1.66 ± 0.11 ^h^
D262R (4/6)	0.11 ± 0.04 ^c^	0.58 ± 0.33	0.34 ± 0.30	2.54 ± 0.25 ^i^
R249D-D262R (4/3)	0.24 ± 0.03	0.30 ± 0.15	0.13 ± 0.09	1.97 ± 0.13

^a^ Significantly different from R249D and D262R (*p* < 0.001). ^b^ Significantly different from R249D (*p* < 0.05) and D262R (*p* < 0.01). ^c^ Significantly different from R249D-D262R (*p* < 0.05). ^d^ Significantly different from R249Q, R249D, R249D-D262R (*p* < 0.0001) and D262R (*p* < 0.05). ^e^ Significantly different from R249Q, R249D-D262R (*p* < 0.0001) and R249D (*p* < 0.01). ^f^ Significantly different from all other samples (*p* < 0.0001). ^g^ Significantly different from R249D (*p* < 0.01). ^h^ Significantly different from D262R (*p* < 0.001). ^i^ Significantly different from R249D-D262R (*p* < 0.05).

## Data Availability

Data are either contained within the article or available in the Appendix A.

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
