# Peer review of "Myosin Transducer Inter-Strand Communication Is Critical for Normal ATPase Activity and Myofibril Structure"

_biology, 2022, doi:10.3390/biology11081137_

Round 1

Reviewer 1 Report

The authors aimed to examine how the R249 mutation of myosin from flight muscle impair myofibril structure and actomyosin ATPase and invitro motility.   The experiments were well conducted.  However, I raised a few questions and comments. 

1) The mutation disrupted the myofibril structure during aging.  This may be due to cellular response (digestion) to small force development but not due to structural instability of myosin filament. This is similar to dystrophy. Please comment in the text.

2) Question related to the above question. Myofibril should be isolated from the young flies.  The author should check the stability in vitro over long time upto 7 days.

3) The authors measured ATPase and in-vitro motility of myosin isolated from mutant Drosophila muscle.  What is the age of Drosophila they isolated from?  In older mutant muscle, myofibril structure was disorganized. Therefore myosin may be damaged. Please comment in the text.

Author Response

1) The mutation disrupted the myofibril structure during aging.  This may be due to cellular response (digestion) to small force development but not due to structural instability of myosin filament. This is similar to dystrophy. Please comment in the text.

The reviewer brings up a significant issue in that lack of force development can lead to muscle atrophy. This has been studied in detail in animals via disuse atrophy models and occurs in humans as well (with similarities to some inherited muscular dystrophy phenotypes). To address this point, we added the following statement and reference to the Discussion section: “It should be noted that the age-related degradation of muscle structure observed in the mutant Drosophila lines may arise from a process akin to disuse atrophy, where reduced force output leads to transcriptional and protein stability changes yielding myofibrillar proteolysis (Jackman et al., 2004).”

Jackman, R.W.; Kandarian, S.C. The molecular basis of skeletal muscle atrophy. Am J Physiol Cell Physiol 2004 287, C834-C843, doi: 10.1152/ajpcell.00579.2003.

2) Question related to the above question. Myofibril should be isolated from the young flies.  The author should check the stability in vitro over long time up to 7 days.

This is an interesting experiment proposed to assess in vitro myofibril stability. However, both senior authors as well as an expert in indirect flight muscle ex vivo mechanics agree that there is no technology available that would allow survival of myofibrils or myofibers from indirect flight muscle in vitro. Hence, we are unable to perform this study. However, we do agree that atrophy of the myofibril structure during aging may arise from its disuse and we have incorporated this information in the text as described in the response to comment number 1 above.

3) The authors measured ATPase and in-vitro motility of myosin isolated from mutant Drosophila muscle.  What is the age of Drosophila they isolated from?  In older mutant muscle, myofibril structure was disorganized. Therefore myosin may be damaged. Please comment in the text.

The reviewer is correct that this is an important issue that was not explicitly addressed in the manuscript. For this work, myosin was isolated from flies that were younger than one day old, to obviate the concern that the reviewer brings up about potential myosin degradation during aging. We now include this information in the Materials and Methods section: “Young flies (3-20 hours post-eclosion) were utilized for myosin isolation to ensure the protein had not been subject to degradation during aging.”

Reviewer 2 Report

The authors of the manuscript «Myosin Transducer Inter-Strand Communication is Critical for Normal ATPase Activity and Myofibril Structure» are recognized experts in the field of molecular mechanisms of muscle contraction. This work continues the series of studies of the relationship between the structure of the myosin molecule and its function. Using a complex of biophysical and biochemical methods, the authors demonstrated that residue R249 located on a β-strand of the transducer of myosin serves as a critical communication link within myosin that controls ATPase and myofibril structure. Very interesting and important results have been obtained.

I have one question for the authors of the manuscript. The flying muscles of Drosophila and the myocardium differ in structural organization. The myocardium is formed by cardiomyocytes, and the flight muscles are formed by fibers. There is a difference in the protein composition of thick filaments, for example, instead of the myosin-binding protein C, in flight muscles, flightin is expressed. The modes of contraction of these muscles are also different. Could these differences, along with the lack of the IHM, be the reason for the different manifestation of the HCM mutation in IFM and the heart? Perhaps because of these differences in the flight muscles of Drosophila, there is a violation of the structure of the sarcomere, and in humans, hypertrophic cardiomyopathy. What can the authors say about this?

Author Response

I have one question for the authors of the manuscript. The flying muscles of Drosophila and the myocardium differ in structural organization. The myocardium is formed by cardiomyocytes, and the flight muscles are formed by fibers. There is a difference in the protein composition of thick filaments, for example, instead of the myosin-binding protein C, in flight muscles, flightin is expressed. The modes of contraction of these muscles are also different. Could these differences, along with the lack of the IHM, be the reason for the different manifestation of the HCM mutation in IFM and the heart? Perhaps because of these differences in the flight muscles of Drosophila, there is a violation of the structure of the sarcomere, and in humans, hypertrophic cardiomyopathy. What can the authors say about this?

We certainly agree with the reviewer as to this point and have now added a statement in the Discussion section that brings it to the reader’s attention: “It should also be noted that while the contractile properties of human cardiac muscle and Drosophila IFM have similarities (Maughan et al., 1998), there are protein differences (for example, human myosin binding protein C and Drosophila flightin), as well as differences in cellular architecture that may play a role in the differential responses of these two muscle types to the R249Q mutation.”  

Maughan, D.; Moore, J.; Vigoreaux, J.; Barnes, B.; Mulieri, L.A. Work production and work absorption in muscle strips from vertebrate cardiac and insect flight muscle fibers. Adv Exp Med Biol 1998,453, 471-480, doi: 10.1007/978-1-4684-6039-1_52.

Reviewer 3 Report

This interesting manuscript describes the results of a comprehensive biochemical, biophysical, locomotory and ultrastructural analysis of the consequences of mutations at R249 of Drosophila melanogaster myosin heavy chain (MyHC). The R249Q mutation in human beta-cardiac MyHC is associated with hypertrophic cardiomyopathy, so there is keen interest in understanding how mutations at this site drive alterations in phenotype. The authors created R249Q, R249D, D262R and R249D-D262R to probe the consequences of the disruption of an R-D salt-bridge at residues 249 and 262 in beta sheets 6 and 7 of the myosin transducer region. The authors conclude that maintenance of the 249-262 salt-bridge is important for myosin ATPase activity, in-vitro actin sliding motility, muscle ultrastructure and function of flight muscles, but the consequences of R249D and D2623R, which presumably maintains the salt bride are not identical to each other. The latter indicates that the residue itself, not the salt-bridge alone, at both sites impacts structure and function. The experiments appear to have been conducted carefully and appropriate controls were included in the study design. The manuscript is very well written and the conclusions of the authors are supported by the results.

I have only a very minor comment. Line 170: “power” should be “powder”.

Author Response

I have only a very minor comment. Line 170: “power” should be “powder”.

This has been corrected as requested.

Round 2

Reviewer 1 Report

The authors aimed to examine how the R249 mutation of myosin from flight muscle impair myofibril structure and actomyosin ATPase and invitro motility.   I raised a few questions and comments.  They answered adequately.

1) The mutation disrupted the myofibril structure during aging.  This may be due to cellular response (digestion) to small force development but not due to structural instability of myosin filament. This is similar to dystrophy. Please comment in the text.                 They described disuse atrophy effect in the Discussion section. I agree.

2) Myofibril should be isolated from the young flies.  The author should check the stability in vitro over long time upto 7 days.

They answered that it is difficult to keep isolated myofibrils for a long time. This may be contamination of protease. I hope that the authors overcome protease contamination and perform in vitro stability experiment in the future. Anyway, I agree.

3) The authors measured ATPase and in-vitro motility of myosin isolated from mutant Drosophila muscle.  What is the age of Drosophila they isolated from?  In older mutant muscle, myofibril structure was disorganized. Therefore myosin may be damaged. Please comment in the text.

The authors isolated them from flies that were younger than one day old. They described this in the Materials and Methods section. I agree.